# Control of transmembrane charge transfer in cytochrome *c* oxidase by the membrane potential

Markus L. Björck[1] & Peter Brzezinski[1]

The respiratory chain in mitochondria is composed of membrane-bound proteins that couple electron transfer to proton translocation across the inner membrane. These charge-transfer reactions are regulated by the proton electrochemical gradient that is generated and maintained by the transmembrane charge transfer. Here, we investigate this feedback mechanism in cytochrome *c* oxidase in intact inner mitochondrial membranes upon generation of an electrochemical potential by hydrolysis of ATP. The data indicate that a reaction step that involves proton uptake to the catalytic site and presumably proton translocation is impaired by the potential, but electron transfer is not affected. These results define the order of electron and proton-transfer reactions and suggest that the proton pump is regulated by the transmembrane electrochemical gradient through control of internal proton transfer rather than by control of electron transfer.

[1] Department of Biochemistry and Biophysics, The Arrhenius Laboratories for Natural Sciences, Stockholm University, SE-106 91 Stockholm, Sweden. Correspondence and requests for materials should be addressed to P.B. (email: peterb@dbb.su.se)

Aerobic respiration involves electron transfer from electron donors such as e.g., NADH, through a series of membrane-bound proteins, to dioxygen that is reduced to water. Part of the free energy released during this electron-transfer process is linked to proton translocation across the membrane, which results in formation of a transmembrane proton electrochemical gradient that is used for ATP production or transmembrane transport.

Transfer of electrons or protons perpendicular to the membrane surface, referred to as electrogenic events, is influenced by the electrical potential across the membrane. In other words, reactions that are involved in generating the potential are influenced by this potential. The focus of the present study is the effect of the electrochemical membrane potential on the reaction of the last component of the electron-transport chain, cytochrome $c$ oxidase (Cyt$c$O), with $O_2$. In this enzyme, electrons are transferred from cytochrome $c$ to the initial electron acceptor, $Cu_A$, and then consecutively to heme $a$ and to the catalytic site composed of heme $a_3$ and $Cu_B$ (for review, see e.g[1–6]). Upon reduction of the catalytic site, $O_2$ binds to heme $a_3$, where the molecule becomes gradually reduced to $H_2O$, which requires a total of four electrons and four protons: $4 e^- + 4 H^+ + O_2 \rightarrow 2 H_2O$. On average one proton per electron transferred to $O_2$ at the catalytic site is pumped from the negative ($n$) to the positive ($p$) side of the membrane (see Fig. 1a, lower). As seen in Fig. 1b, $Cu_A$ is located near the $p$ side, while heme $a$ and the catalytic site are found within the membrane-spanning part of the protein. Hence, electrogenic reactions in Cyt$c$O are electron transfer from $Cu_A$ to heme $a$, proton uptake from the $n$ side to the catalytic site as well as proton pumping[7–9].

Results from earlier studies with Cyt$c$O reconstituted in lipid vesicles have shown that in the presence of a transmembrane electrochemical gradient (positive on the outside) the Cyt$c$O turnover is slowed typically by a factor of 5–20[10–12]. The effect is smaller with sub-mitochondrial particles (SMPs) where values in the range 1.4–3.5 were reported, depending on the used electron donor as well as other experimental conditions[13,14].

In the present study we have investigated the effect of a transmembrane electrochemical potential on specific electron and proton-transfer events during reaction of the reduced Cyt$c$O with $O_2$ in SMPs. The potential was generated by addition of ATP, which initiates proton pumping by ATP-synthase. In this system the buildup of the electrical potential acts to lower the available free energy for proton pumping by Cyt$c$O (Fig. 1a, upper).

Each Cyt$c$O molecule was reduced by four electrons, one at each of the metal co-factors, $Cu_A$, heme $a$, heme $a_3$ and $Cu_B$. At a specific time after addition of ATP, the reaction of Cyt$c$O with $O_2$ was initiated synchronously in the entire Cyt$c$O population by flash-induced dissociation of carbon monoxide from the reduced Cyt$c$O in the presence of $O_2$. The step-wise oxidation of the Cyt$c$O was followed in time by monitoring absorbance changes associated with oxidation of hemes $a$ and $a_3$. The reaction sequence is described in Fig. 1b. Briefly, the dissociation of CO yields the reduced Cyt$c$O (called **R**), which binds $O_2$ to the heme $a_3$ iron forming a state that is called **A** with a time constant of ~10 μs at 1 mM $O_2$ (e.g[15]). The O–O bond is broken and a ferryl state, called $\mathbf{P_R}$, $\left[ Fe^{4+}_{a_3} = O^{2-} \right] \left[ Cu^{2+}_B - OH^- \right]$ is formed at the catalytic site[16–18] simultaneously with oxidation of heme $a$[19–21] and proton transfer from a nearby Tyr residue (not explicitly drawn in the Fig. 1b) with a time constant of ~30 μs. Next, a proton is taken up from solution to the catalytic site leading to formation of a ferryl state called **F**, $\left[ Fe^{4+}_{a_3} = O^{2-} \right] \left[ Cu^{2+}_B - OH_2 \right]$, with a time constant of 100 μs at pH 7[20,22,23]. In addition, the electron at $Cu_A$

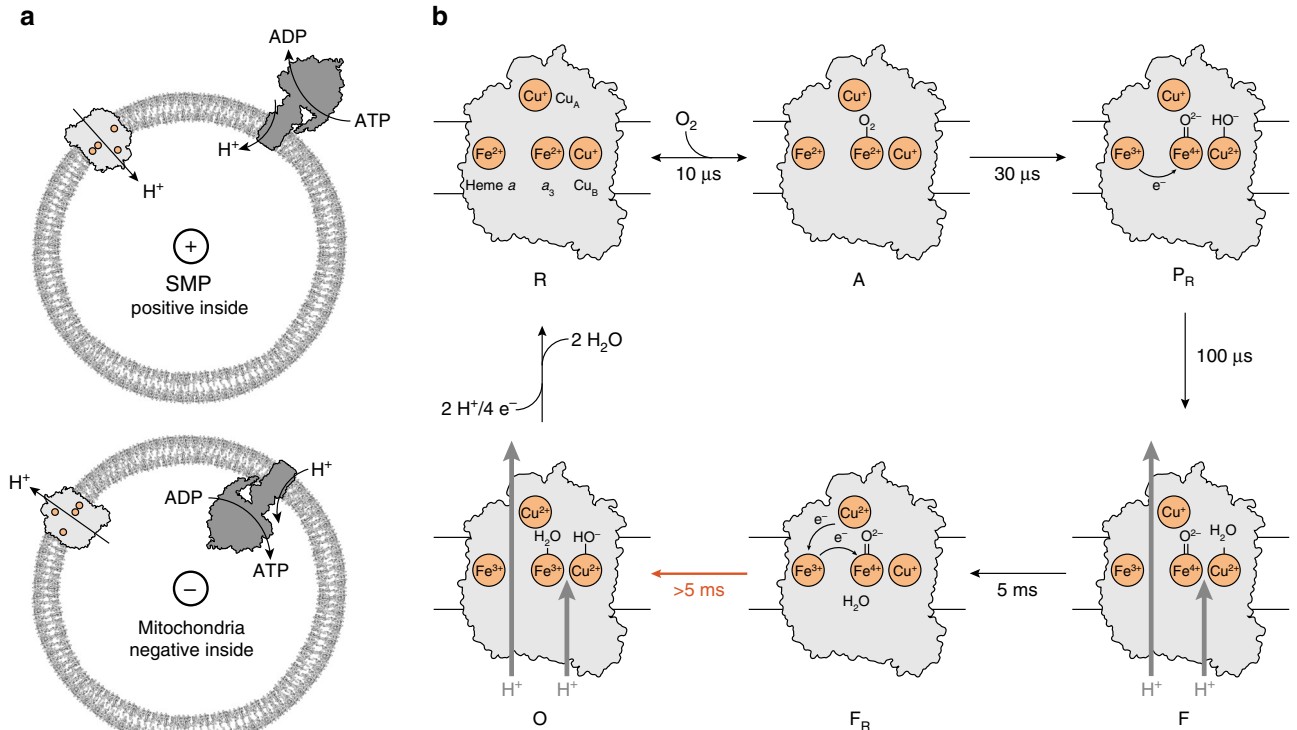

**Fig. 1** Experimental system and model. **a** Cyt$c$O and ATP-synthase orientation in SMPs and mitochondria, respectively. **b** The reaction studied in this work. Upon initiation of the reaction by a light flash CO dissociates to form state **R**. Then, $O_2$ binds to the reduced Cyt$c$O after which the reaction proceeds as shown and described in the text. The data from this work suggest that the $\mathbf{F_R} \rightarrow \mathbf{O}$ step is slowed/blocked by the electrochemical proton gradient (red arrow). The release of $H_2O$ is indicated in the $\mathbf{O} \rightarrow \mathbf{R}$ step, but one of the $H_2O$ molecules could be released earlier in the cycle. One of the protons taken up in this reaction is bound by a Tyr residue that is not explicitly drawn in the figure (the Tyr also donates a proton upon forming $\mathbf{P_R}$)

equilibrates with heme $a$, but in the SMPs this equilibrium is shifted toward $Cu_A$ (see below). After transfer of an electron and a proton to the catalytic site, the oxidized state, $\left[Fe_{a_3}^{3+} - OH_2\right]\left[Cu_B^{2+} - OH^-\right]$, $\mathbf{O}$, is formed over a millisecond time scale (in this work a 5-ms component, see below). Typically the $\mathbf{F} \rightarrow \mathbf{O}$ reaction is observed as a single kinetic component, but results from theoretical and experimental studies indicate that it may be composed of two separable events: first electron transfer from $Cu_A$/heme $a$ to $Cu_B$, yielding a state that is called $\mathbf{F_R}$[24–28] and then proton uptake to the catalytic site to yield state $\mathbf{O}$ (see Fig. 1b).

The data from the present study show that the last step of the reaction, $\mathbf{F} \rightarrow \mathbf{O}$, was influenced by membrane potential. Furthermore, the data indicate that electron transfer from heme $a$ to the catalytic site was unaffected by the membrane potential, while proton uptake and presumably pumping were slowed. As a result, state $\mathbf{F_R}$ was formed before proton uptake and pumping. This scenario suggests that CytcO turnover is regulated by altering the rate of proton transfer rather than electron transfer. Furthermore, the data define the order of electron and proton-transfer reactions during CytcO turnover.

## Results

**Orientation of the respiratory chain in the SMPs**. Results from earlier studies have shown that the orientation of the protein components of the SMPs is inverted as comparted to that of the native inner mitochondrial membrane[29,30] (see Fig. 1a). To determine the orientation of the respiratory chain in our preparation we added consecutively NADH and dithionite under anaerobic conditions and compared the heme reduction levels after the additions. Because NADH does not penetrate the membrane it reduces only the redox components of the respiratory chain in SMPs with an inverted orientation. Dithionite reduces all components irrespectively of orientation. A very small further reduction was observed after addition of dithionite to the NADH-reduced membranes (Supplementary Figure 1), indicating that essentially all intact SMPs had their NADH-binding sites on the outside. Consequently, also the $F_1$ part of the ATP-synthase was oriented to the outside of the SMPs as shown in Fig. 1a.

**Formation of membrane electrochemical potential**. We first studied formation of a transmembrane electrical potential ($\Delta\Psi$) and proton concentration gradient ($\Delta pH$) in the SMPs upon addition of ATP. Figure 2a shows changes in absorbance at 623 nm, $\Delta A^{623}$, as a function of time after addition of ATP to SMPs in the presence of the membrane potential-sensitive dye oxonol VI[31]. As seen in the figure, the absorbance increased, which indicates that the potential increased. The initial increase was transient after which it decreased over a time scale of ~200 s to reach a slow decay slope. The inset shows a similar experiment, but done in a stopped-flow apparatus with a higher time resolution. In the inset it is seen that the maximum of the transient potential change was observed after ~3 s. Upon addition of nigericin, which equilibrates the potassium and proton concentration gradients to convert the $\Delta pH$ component into a $\Delta\Psi$ (in the current experiment where protons are continuously pumped by the ATP synthase), the absorbance increased further (Fig. 2a). No changes in absorbance were observed in the presence of valinomycin, which abolishes $\Delta\Psi$ but not $\Delta pH$. A calibration curve done with liposomes of the same size as the SMPs is shown in Supplementary Figure 2. Based on these data we estimated the membrane electrical potential established across the SMPs to be ~100 mV and the $\Delta pH$ at the time of nigericin addition to be ~0.4 pH units (see Fig. 2a).

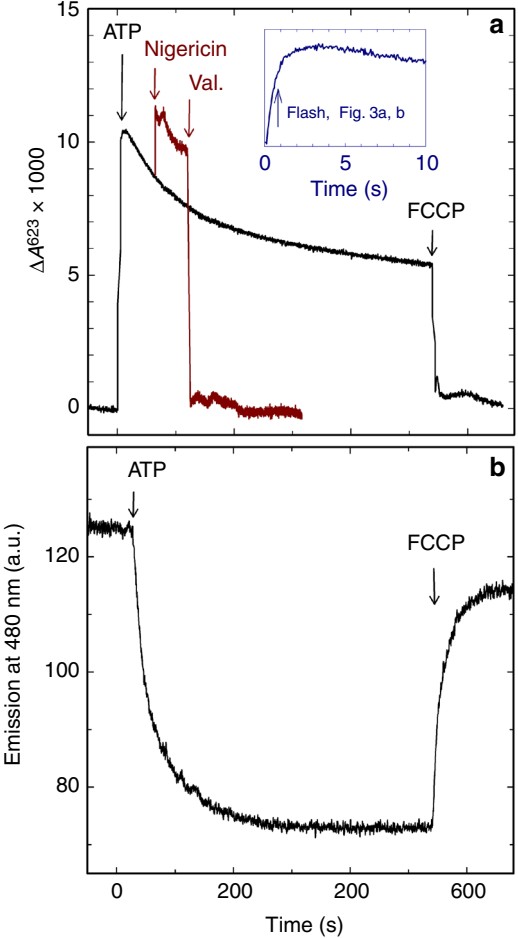

**Fig. 2** Generation of an ATP-induced transmembrane electrochemical potential in SMPs. **a** The membrane electrical potential ($\Delta\Psi$) was monitored using the dye oxonol VI. The absorbance changes at 623 nm were measured as a function of time. ATP was added to the SMPs solution, containing oxonol VI, at $t = 0$ (black trace). The signal increased, consistent with proton pumping into the SMPs. Addition of nigericin after ~60 s (red trace) resulted in an increase in absorbance as the proton gradient ($\Delta pH$) was converted into an electrical potential. In the presence of valinomycin, (val., added after ~120 s) the absorbance decreased to the same level as that before addition of ATP (red trace). Addition of FCCP (at ~540 s) removes both $\Delta\Psi$ and $\Delta pH$ (black trace). Mixing artifacts caused by the additions were removed for clarity. The absorbance changes were also resolved on a shorter time scale using a stopped-flow device (see inset in **a**). The arrow indicates the time at which the reaction was started in the experiments shown in Fig. 3a, b. Experimental conditions: 0.1 mg SMPs in 1 ml buffer, 2 µM oxonol VI, 1 µM valinomycin. The final ATP concentration was 160 µM. In the stopped-flow experiment it was 1 mM. **b** Proton pumping to the interior of the SMPs (initiated by addition of ATP at $t = 0$) was monitored using the fluorescent dye ACMA. As the inside of the SMPs became acidified the emission of the fluorescent dye was quenched. Experimental conditions: 0.1 mg SMPs in 1 ml buffer, 200 nM ACMA, the final ATP concentration was 160 µM. The fluorophore was excited at 410 nm and the emission was recorded at 480 nm. (a.u.) is arbitrary units

Changes in the transmembrane pH gradient were monitored by measuring fluorescence changes of the dye 9-amino-6-chloro-2-methoxyacridine (ACMA) (Fig. 2b). The fluorescence decreased upon proton pumping into the SMPs[32] over a time scale of ~200 s, i.e., the same time scale as the initial fractional decay in $\Delta\Psi$. Addition of the proton ionophore FCCP eliminated the fluorescence signal.

The initial transient increase in membrane potential was much faster (~3 s) than formation of the proton concentration gradient (~100 s), which is consistent with results from earlier studies[31]. This difference in time scales is observed because formation of $\Delta\Psi$ requires transfer of only a few charges across the membrane (charging of a spherical lipid-membrane capacitor), while formation of $\Delta pH$ requires a large number of turnovers by the ATP synthase. The decay in the membrane potential, starting at ~5 s after addition of ATP, is presumably caused by equilibration of proton pumping and ion leaks being established across the membrane.

We also measured the $O_2$-reduction rate upon addition of succinate to the SMPs in the presence and absence of the proton ionophore FCCP. The ratio of these rates is defined as the respiratory control ratio (referred to as RCR). We obtained an RCR value of $1.30 \pm 0.05$ (SD of 10 measurements).

**Reaction with $O_2$.** We reduced the SMPs with ascorbate and incubated the sample under an atmosphere of CO, which resulted in formation of the reduced CytcO–CO complex. This sample solution was mixed 1:1 with an $O_2$-saturated (~1.2 mM $O_2$) solution containing ATP in a stopped-flow apparatus. After a time delay of ~0.8 s the CO ligand was dissociated by means of a laser flash, which allowed $O_2$ to bind to initiate the reaction of the reduced CytcO with $O_2$. During the time window of mixing and initiation of the reaction, an electrical potential was established across the membrane, but the proton concentration gradient was expected to be small after ~0.8 s (c.f. Fig. 2a, b).

The reaction was monitored at two wavelengths, 445 nm and 605 nm. At 445 nm hemes $a$ and $a_3$ contribute by ~40% and ~60%, respectively, to the total reduced-minus-oxidized difference spectrum[33]. At 605 nm the main contribution is from redox changes at heme $a$, which contributes by ~80% of the total change[33,34].

At 445 nm the absorbance increased at the time of the laser flash ($t = 0$, Fig. 3a), which is associated with dissociation of the CO ligand. The absorbance level immediately after illumination corresponds to that of the reduced CytcO (state **R**, see scheme in Fig. 1b). The decrease in absorbance at both 445 nm and 605 nm (Fig. 3b) is associated with $O_2$ binding to heme $a_3$ forming state **A** ($\tau \cong 20$ μs at 0.5 mM $O_2$) and formation of the peroxy state, $P_R$, which is associated with electron transfer from heme $a$ to the catalytic site ($\tau \cong 30$ μs).

In the next step a proton is transferred to the catalytic site to form the ferryl, **F** state with a time constant of ~100 μs at pH 7. The $P_R \rightarrow F$ reaction is associated with pumping of one proton across the membrane and occurs over the same time scale as a shift in the electron equilibrium from $Cu_A$ toward heme $a$. Results from earlier studies have shown that this electron transfer yields an absorbance increase or plateau in the time range approximately 200–500 μs at 445 nm, depending on the fraction reduced heme $a$[20]. With the bovine CytcO in detergent solution a relatively large increase in absorbance was seen[20], but in the SMPs we observed only a plateau, indicating a small fractional electron transfer from $Cu_A$ to heme $a$, also without a membrane potential (see Fig. 3a). In the last step of the reaction the fourth electron is transferred from heme $a$ to the catalytic site, which yields the oxidized CytcO (state **O**). The $F \rightarrow O$ transition ($\tau \cong 5$ ms) is linked to proton uptake to the catalytic site and pumping across the membrane, which are both electrogenic. The time constant of the reaction is typically ~1 ms in detergent solution[35], but in the SMPs it was slowed to ~5 ms (Fig. 3AB), as observed previously for membrane-bound CytcO[36].

The amplitude of the absorbance change at 445 nm decreased by $14 \pm 1\%$ (SD, 3 measurements) in the presence of a membrane

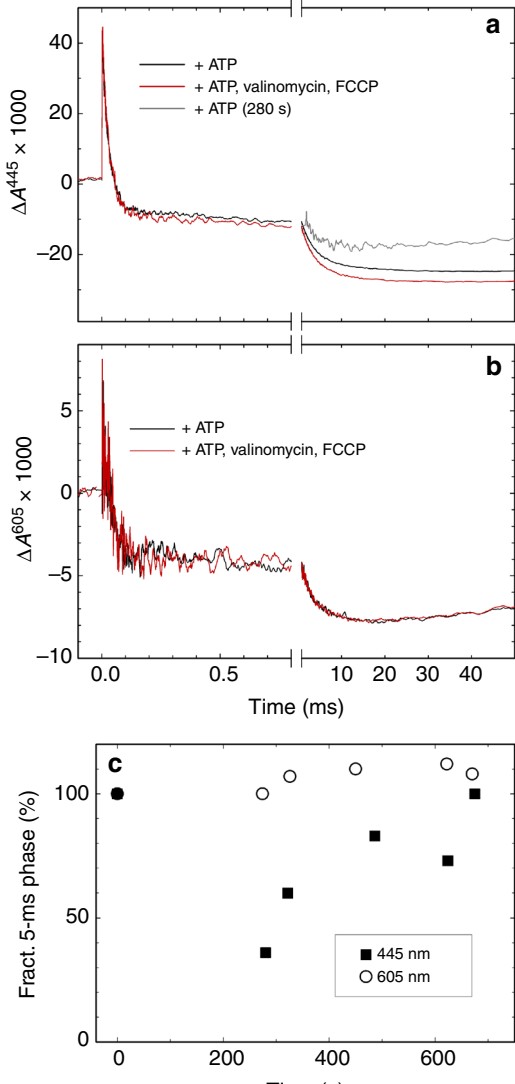

**Fig. 3** Absorbance changes associated with the reaction of CytcO with $O_2$ in sub-mitochondrial particles. The reduced CytcO was mixed with an $O_2$-saturated solution containing ATP. After 0.8 s the reaction of the CytcO with $O_2$ was initiated by a laser flash (at $t = 0$ in the graph). It was monitored at 445 nm (**a**), which reflects the redox states of hemes $a$ and $a_3$, and at 605 nm (**b**), which reflects the redox state mainly of heme $a$. Experimental conditions after mixing: 250 mM sucrose, 50 mM KCl, 10 mM phosphate buffer at pH 7.4, 5 mM MgCl$_2$, 0.1 mM EDTA, 25 mM ATP, 1.9 mg ml$^{-1}$ SMPs with or without 10 μM valinomycin and 200 nM FCCP. The mixing ratio was 1:1. The grey graph is the 280-s time point from **c**. Typically, 10 or 20 traces were averaged at 445 nm and 605 nm, respectively. In **b** a laser artifact has been truncated. **c** Amplitude of the 5-ms component as a function of time monitored at 445 nm (filled symbols) and 605 nm (open symbols). ATP was added to the anaerobic SMP solution before the sample was transferred to the stopped-flow apparatus. The first time point in this graph was obtained before addition of ATP (set to 100%). The errors, estimated from the noise level of the traces, were typically smaller than the marks. Experimental conditions before mixing: 250 mM sucrose, 50 mM NaCl, 20 mM Hepes pH 7.4, 5 mM MgCl, 1 mM ATP and 5.5 mg ml$^{-1}$ SMP. The SMP:O$_2$ solution mixing ratio was 1:5. The time delay between mixing and laser flash was 0.2 s

potential (Fig. 3a). At 605 nm the difference in the amplitudes without and with a membrane potential for the 5-ms component was <2% (3 measurements) (Fig. 3b). A comparison of the absorbance changes at these two wavelengths indicates that with a membrane potential, electrons remained at the catalytic site at the end of the reaction (after ~10 ms), while heme $a$ was essentially fully oxidized independently of the membrane potential.

The experiments described above were carried out in a stopped-flow apparatus where ATP and $O_2$ were added at the same time. Therefore, the delay time between addition of $O_2$ (and thus ATP) and flash-induced initiation of the reaction must be kept short enough to prevent $O_2$ from reacting before flash photolysis of the CO ligand. To increase the time between ATP addition and start of the reaction, we also carried out an experiment in which ATP was added to the SMP-containing cuvette in the absence of $O_2$. The sample was then introduced into the stopped-flow apparatus and mixed with $O_2$ as described above. Because it took ~5 minutes for the anaerobic transfer of the sample to reach the cuvette, the first time point after addition of ATP was measured after ~280 s (Fig. 3c). As seen in Fig. 3a (grey trace), at this time the amplitude of the 5-ms component in the absorbance change had decreased to ~50% of its maximum value (100% was set as the amplitude before ATP addition). This drop in amplitude is larger than that observed after 0.8 s incubation (see above) (Fig. 3a) presumably because at 280 s the pH gradient was fully developed (c.f. Figure 2). At times >280 s the amplitude increased again to reach ~100% at ~600 s, presumably when $\Delta\Psi$ decreased (see slow decrease in absorbance at times >280 s, Fig. 2a), while $\Delta$pH remained at a constant level (see Fig. 2b). As seen in Fig. 3c, the amplitude of the 5-ms component at 605 nm remained at an approximately constant level during the same time period after addition of ATP.

## Discussion

We have investigated the effect of membrane potential on the reaction of the four-electron reduced Cyt$c$O with $O_2$ in SMPs. No effect of membrane potential on $\mathbf{P_R}$ formation was observed ($\tau \cong 30\ \mu s$). This reaction step involves electron transfer from heme $a$ to the catalytic site, which occurs parallel to the membrane surface. Hence, this electron transfer is expected to be insensitive to the external potential. The following two reaction steps are linked to proton uptake and pumping, and electron transfer perpendicular to the membrane plane; $\mathbf{P_R} \rightarrow \mathbf{F}$ and $\mathbf{F} \rightarrow \mathbf{O}$ (see Fig. 1b). Results from earlier studies showed that both these steps are electrogenic[8,37], which suggests that both reactions could, in principle, be sensitive to an external potential.

We discuss the last step of $O_2$ reduction to $H_2O$, the $\mathbf{F} \rightarrow \mathbf{O}$ reaction, first. It involves simultaneous electron transfer from the $Cu_A$-heme $a$ equilibrium to the catalytic site, proton uptake to the catalytic site and proton pumping. In other words, the $\mathbf{F} \rightarrow \mathbf{O}$ reaction is representative of each step in Cyt$c$O turnover where electrons are delivered one-by-one from cytochrome $c$, via Cyt$c$O, to $O_2$. The electron and proton-transfer reactions in $\mathbf{F} \rightarrow \mathbf{O}$ occur over the same time scale and are presumably rate-limited by the release of the pumped proton[38–40].

At 605 nm the absorbance changes with and without electrochemical potential were nearly identical over a time scale of ~10 ms (Fig. 3b and c), which shows that the extent of heme $a$ oxidation during the time of the $\mathbf{F} \rightarrow \mathbf{O}$ reaction was independent of the membrane potential. However, the amplitude of the absorbance change at 445 nm decreased in the presence of membrane potential (Fig. 3a), which qualitatively indicates less oxidation of the catalytic site. Collectively, these observations indicate that all four electrons are transferred to $O_2$ at the catalytic site over a time

scale of 5 ms, but that this electron transfer does not lead to complete reduction of the bound $O_2$ to $H_2O$, i.e., that proton uptake to form $H_2O$ is impaired.

As discussed in the Introduction section, the reaction steps that are part of the $\mathbf{F} \rightarrow \mathbf{O}$ reaction have been considered in detailed theoretical studies[24,25]. It was suggested that over the time scale of $\mathbf{F} \rightarrow \mathbf{O}$ reaction, the electron is initially transferred to $Cu_B^{2+}$ yielding state $\mathbf{F_R}$[26] (see Fig. 1b), which was also indicated from experiments[27,28]. As already noted above, electron transfer from heme $a$ to $Cu_B$ occurs parallel to the membrane surface, and it is therefore expected to be insensitive to the membrane potential. Reduction of $Cu_B$ is followed in time by proton uptake to complete $O_2$ reduction to $H_2O$ and proton pumping across the membrane. In principle, the proton-pumping event could occur over the time scale of either the $\mathbf{F} \rightarrow \mathbf{F_R}$ or the $\mathbf{F_R} \rightarrow \mathbf{O}$ reaction, but it is more likely to coincide with the latter in analogy with proton pumping during the $\mathbf{P_R} \rightarrow \mathbf{F}$ reaction, i.e. the previous step[36,41] (see Fig. 1b). Furthermore, we note that upon one-electron reduction of Cyt$c$O in the $\mathbf{F}$ state, the $\mathbf{F} \rightarrow \mathbf{O}$ reaction was found to be biphasic with a second, slower component accounting for 3/4 of the total electrogenicity[28]. These two components would be associated with electron ($\mathbf{F} \rightarrow \mathbf{F_R}$) and proton ($\mathbf{F_R} \rightarrow \mathbf{O}$) transfer (as well as pumping), respectively. Because proton uptake and pumping have trajectory components that are perpendicular to the membrane surface, the driving force ($-\Delta G$) of these reactions is expected to be smaller with than without a membrane potential, i.e. the $\mathbf{F_R} \rightarrow \mathbf{O}$ reaction would be impaired.

As pointed out above, we assume that in the absence of membrane potential the two reaction steps, $\mathbf{F} \rightarrow \mathbf{F_R}$ and $\mathbf{F_R} \rightarrow \mathbf{O}$, are inseparable in time, presumably because decay of state $\mathbf{F_R}$ to $\mathbf{O}$ is faster than its formation. This assumption also explains why the time constant of the $\mathbf{F} \rightarrow \mathbf{F_R}$ reaction with a transmembrane potential is the same as that of the $\mathbf{F} \rightarrow \mathbf{O}$ reaction without a membrane potential; the time constant for electron transfer from heme $a$ to $Cu_B$ to form $\mathbf{F_R}$ is not affected by the membrane potential, while the next reaction step, $\mathbf{F_R} \rightarrow \mathbf{O}$, is slowed/impaired such that the state formed over the time scale of measurement is $\mathbf{F_R}$. Furthermore, the ~14% decrease of the absorbance amplitude at 445 nm upon formation of a membrane potential would reflect the contribution of $\mathbf{F_R}$, relative to that of $\mathbf{O}$.

Assuming the scenario outlined above, the amplitude of the 5-ms component in the presence of membrane potential would reflect electron transfer from heme $a$ to $Cu_B$, i.e., mainly oxidation of heme $a$, for which the absorption coefficient at 445 nm is ~60 mM$^{-1}$cm$^{-1}$[33,34]. The absorption coefficient for state $\mathbf{F}$ (relative to the oxidized Cyt$c$O) is ~10 mM$^{-1}$cm$^{-1}$[7]. Thus, if $\mathbf{F}$ with an oxidized heme $a$ was the final state after 5 ms, we would expect a decrease of the 5-ms component amplitude by ~15%. This number is approximately equal to the observed 14% change in amplitude. However, the absorption coefficient for state $\mathbf{F_R}$ may differ from that of state $\mathbf{F}$ because of the additional electron at $Cu_B$ in the former, which may explain why we observed an even bigger decrease in the 445-nm amplitude for the data shown in Fig. 3c. The contribution of state $\mathbf{F}$ at 605 nm is ~5% of that of heme $a$[7,42], which explains why the absorbance levels were the same after 5 ms at this wavelength.

The conclusion from the discussion above is that in the presence of an electrochemical potential the Cyt$c$O state reached after the 5-ms component is $\mathbf{F_R}$. In other words, the decrease in the amplitude of this component is a reflection of a considerable decrease in the rate of the next reaction step, i.e., $\mathbf{F_R} \rightarrow \mathbf{O}$. It is difficult to estimate the decay time constant of this state in the SMPs because after the Cyt$c$O is oxidized it becomes slowly re-reduced by ascorbate that must be present in the experiment for preparation of the reduced form of the enzyme. Nevertheless, as

seen in Fig. 3a the absorbance level was essentially unaltered for about 50 ms, which defines the lower limit for the life time of $F_R$.

We now turn to the $P_R \rightarrow F$ reaction. The absorbance changes associated with this reaction are very small, also at the peak wavelength of ~580 nm[7] and could not be resolved here with the scattering SMPs. As discussed above, the $P_R \rightarrow F$ reaction, and proton translocation that is linked to the reaction[36,40,43] occur over the same time scale as a shift in the equilibrium between $Cu_A$ and heme $a$, leading to a fractional reduction of heme $a$[44]. This shift yields absorbance changes, for example, at 445 nm and at 605 nm, associated with reduction of heme $a$. However, the extent of heme $a$ reduction varies between CytcOs from different species[20] and depending on experimental conditions (detergent, lipids, pH etc.) (e.g[22]). For the SMPs studied here the extent of the $Cu_A$—heme $a$ electron transfer was small during the $P_R \rightarrow F$ reaction (Fig. 3, also indicated in Fig. 1b). We did not observe any differences in the absorbance changes with or without a membrane potential over the time scale of the $P_R \rightarrow F$ reaction, presumably because the membrane potential established by ATP hydrolysis acts to shift the $Cu_A$—heme $a$ electron equilibrium even further toward $Cu_A$[11]. It should be noted that the proton uptake and pumping during $P_R \rightarrow F$ are independent of the electron transfer from $Cu_A$ to heme $a$[44]. Because heme $a$ was oxidized over a time scale of ~5 ms (see Fig. 3b), the $P_R \rightarrow F$ reaction must occur with a time constant $\leq 5$ ms, i.e. it is not rate limiting for the overall oxidation of CytcO.

Data from earlier studies of the $P_R \rightarrow F$ reaction in the $R.$ $sphaeroides$ CytcO in $H_2O$ and $D_2O$, respectively, showed that the ratio of the proton-release rates was ~7 (defined as kinetic isotope effect, KIE)[38]. In contrast, the KIE of the $P_R \rightarrow F$ chemical reaction and the associated proton-uptake reactions was 2-3[45]. These differences in the KIEs yielded a delay of the proton release, relative to proton uptake in $D_2O$. This means that even if the release of the pumped proton would be slowed or impaired by the membrane potential, the $F$ state could be formed at a rate that is independent of the membrane potential.

We measured a RCR of ~1.3, which is at the lower end of the range of previously reported values[13,14]. Nevertheless, the value is significantly larger than unity and both the electrical component and pH gradient remained stable during the time of the measurements (c.f. Figure 2). Furthermore, we note that in the earlier studies of the RCR oligomycin was used to block proton leaks via the ATP synthase. Because in the current study we used the ATP synthase to maintain a transmembrane electrochemical potential, oligomycin had to be excluded.

The mechanistic scenario suggested from this work defines the order of electron and proton transfer during $O_2$ reduction at the CytcO catalytic site. Electron transfer from heme $a$, $F \rightarrow F_R$, occurs first and is insensitive to the membrane potential. This electron transfer may be linked to proton uptake from the $n$ side of the membrane to a proton-loading site (PLS)[43,46,47], but not to the catalytic site. Uptake of a proton to the catalytic site, $F_R \rightarrow O$, and release of the pumped proton is significantly slowed by the membrane potential. Alternatively, only the electron transfer from heme $a$ to the catalytic site ($F \rightarrow F_R$), parallel to the membrane surface, is insensitive to the membrane potential, in agreement with a model proposed earlier[36,41]. Both scenarios would imply that electron transfer to the catalytic site can occur without accompanying proton uptake, that release of the pumped proton from PLS is mechanistically linked to the proton uptake to the catalytic site and that the latter reaction is influenced by the electrochemical proton gradient. We note that the electrochemical potential created in the SMPs was somewhat lower than that found in the native system, i.e., the effects were observed at potentials lower than that found in vivo.

The data from these studies suggest a more general mechanism by which the cell may regulate the rate of respiration. It has been shown that CytcO is a key site for regulation of the OXPHOS[48]. Even though there are several modes of regulation of the enzyme (see e.g[1,48–51]) modulation of the turnover rate by changes in the proton electrochemical potential is a general principle. The data from the present study indicate that the CytcO turnover is regulated by slowing proton uptake and presumably pumping in the $F_R \rightarrow O$ reaction. Because during CytcO turnover, formation of $O$ is followed in time by re-reduction, also electron transfer would be slowed by the membrane potential. This mode of regulation is consistent with the three-dimensional arrangement of the proton pathways and metal cofactors in CytcO[52–55]. Proton uptake is perpendicular to the membrane surface and takes place across 2/3 of the transmembrane distance, across a hydrophobic region of the CytcO where the dielectric constant is expected to be small. Proton pumping spans across the entire distance of the membrane dielectric. Hence, the driving force of the proton-transfer reactions is expected to be sensitive to the transmembrane potential. Electron transfer from $Cu_A$ to heme $a$, on the other hand, takes place across only 1/3 of the transmembrane distance, through a region of the protein that is likely to have a relatively large dielectric constant because of the large amount of water molecules[52–55]. In conclusion, the data suggest that the activity of CytcO is regulated by the external electrochemical potential as a result of altering the driving force for proton transfer in the very last step of $O_2$ reduction to $H_2O$.

## Methods

**Reagents**. All chemicals used in this study were of the purest grade available and purchased from Sigma Aldrich.

**Preparation of bovine heart mitochondria**. Bovine heart mitochondria were prepared as described in[56]. Procedure three was used with the following modifications: 600 ml sucrose buffer was used in the blender step and 45 s pulses were used instead of 15 s, the homogenate was centrifuged at 1600x$g$ for 15 min in the first centrifugation step, and at 10,000x $g$ for 30 min in the following steps. The mitochondria were separated into light and heavy fractions as described[56]. Briefly, the sample containing mitochondria was centrifuged at 10,900x $g$. The pellet consisted of two parts, one light, slightly less colored and one darker more firm pellet. The lighter fraction (broken mitochondrial fragments) was removed by decantation. The procedure was then repeated until only the heavy fraction of mitochondria was present after centrifugation.

**Preparation of SMPs**. A sample containing the heavy fraction bovine heart mitochondria was diluted to ~3.3 mg ml$^{-1}$ in a buffer containing 250 mM sucrose, 50 mM KCl, 10 mM phosphate-buffer at pH 7.4, 5 mM MgCl$_2$ and 0.1 mM EDTA, and the solution was homogenized in a glass potter. The mitochondrial suspension was then sonicated for 1 min at maximum output using a Vibra cell sonicator (Sonics & Materials Inc.) while cooled using a mixture of ice and water. The sonicated solution was diluted 1:1 in sucrose buffer and then centrifuged at 14,000x $g$ for 10 min to remove debris and intact mitochondria. The pellet was discarded and the supernatant was centrifuged at 10,5000x $g$ for 1 h. The supernatant was discarded and the pellet was dissolved in the same buffer as previously to a concentration in the range 10–20 mg ml$^{-1}$. The SMPs were flash-frozen in liquid nitrogen and stored at −80 °C until use.

**Flow-flash measurements**. The SMPs were dissolved in the sucrose buffer described above to a concentration of ~3.8 mg ml$^{-1}$. The sample was loaded into a Thunberg cuvette and air was exchanged for nitrogen. The sample was then reduced with 4 mM ascorbate and 1 μM PMS. After the sample was fully reduced, nitrogen was exchanged for CO. The absorbance changes associated with reduction and binding of CO to CytcO were monitored using a Cary 4000 spectrophotometer (Agilent).

The flow-flash measurements were performed essentially as described previously[57,58] using a laser flash-photolysis system combined with a stopped-flow apparatus (Applied Photophysics). The mixing ratio of sample and saturated oxygen buffer was 1:1 and the cuvette path length was 1.00 cm. The delay time between mixing and laser flash was 0.8 s. The oxygen-containing buffer consisted of the sucrose buffer used to dissolve SMPs and it also contained either only ATP or ATP, 200 nM FCCP and 10 μM valinomycin. The ATP concentration after mixing was 25 mM.

**Measurement of membrane potential and proton pumping**. The membrane potential generated upon addition of ATP to SMPs was monitored using the potential-sensitive dye Oxonol VI (see[59]). SMPs were dissolved in sucrose buffer (see above) at a concentration of 0.1 mg ml$^{-1}$. Changes in absorbance were monitored at 623 nm using a Cary 100 spectrophotometer (Agilent). The dye oxonol VI was then added to a final concentration of 2 μM. Nigericin was added (1 μM) to convert the pH gradient into potential under the conditions of these measurements. As a control, valinomycin was added and finally FCCP (1 μM final concentration of each) to dissipate the membrane potential and proton gradient. The final volume after the additions was 1 ml and final ATP concentration was 160 μM.

A stopped-flow apparatus, equipped with a diode-array detector (Applied Photophysics), was used to increase the time resolution of the potential measurement. A sample containing SMPs in sucrose buffer, supplemented with Oxonol VI (2 μM after mixing) was mixed 1:1 with a sucrose buffer containing ATP (1 mM after mixing). Changes in absorbance were monitored at 623 nm (relative to those measured at 604 nm). The SMP concentration was 0.1 mg ml$^{-1}$ after mixing.

The oxonol VI response to the membrane potential was calibrated using asolectin liposomes of the same size as the SMPs. The liposomes were prepared from washed soybean lipids (L-α-phosphatidylcholine, type II-S) essentially as described in[60]. The preparation buffer contained 20 mM Hepes pH 8, 150 mM NaCl and 0.5 mM KCl. A solution of asolectin (total 60 mg) was prepared at 40 mg ml$^{-1}$ and 2% sodium cholate was added. The lipids were sonicated for 4 min in 15 s on and 15 s off cycles using a Vibra cell sonicator (Sonics & Materials Inc.) with the amplitude set to 35% of the maximal. After formation of the liposomes detergent was removed by three-step dialysis in the same buffer, but without detergent (200 ml for 3 h, 300 ml for 3 h and 500 ml overnight).

The calibration was performed by gradually increasing the external KCl concentration in the presence of valinomycin and monitoring the associated absorbance changes of the dye at 623 nm. The membrane potential associated with the dye response to the potassium gradient was calculated using the Nernst equation. The lipid concentration was 8 mg ml$^{-1}$, 2 μM oxonol VI and 1 μM valinomycin was used. External potassium concentrations were (in units of mM): 2.5, 5, 10, 20, 40 and 50 (0.5 mM KCl inside). The buffer composition was the same as that used for liposome preparation and the final volume was 1 ml.

To measure changes in proton concentration, the SMPs were dissolved in a sucrose buffer to 0.1 mg ml$^{-1}$ and the fluorescent dye ACMA[61] was added (200 nM). The sample was excited at 410 nm and emission was recorded at 480 nm. A baseline was recorded for ~60 s and then proton pumping was initiated by addition of ATP (160 μM final concentration). Measurements were performed using a fluorometer (Cary Eclipse fluorescence spectrophotometer, model G9800 A, Agilent).

**Respiratory control ratio**. The RCR was determined by measuring the oxygen-reduction rate, using a Clark-type electrode (Hansatech Instruments), upon addition of the SMPs, in the absence and presence of the proton ionophore FCCP. All measurements were performed in the buffer used to dissolve the SMPs (see Preparation of SMPs above), with the addition of fatty-acid free BSA (1 mg ml$^{-1}$). Succinate was added to the measurement chamber (final concentration 10 mM) and a background was recorded. SMPs, pre-incubated with 10 mM succinate on ice for 15 min to activate complex II, were then added (final concentration 0.09 mg ml$^{-1}$) and the oxygen reduction was monitored before and after addition of FCCP at 1 μM final concentration.

**Data availability**. Data supporting the findings of this manuscript are available from the corresponding author upon reasonable request.

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

## Acknowledgements
We would like to acknowledge Drs. Pia Ädelroth, Christoph von Ballmoos and Irina Smirnova for valuable advice and discussions. We would also like to thank Jacob Schäfer for help in preparation of Fig. 1. These studies were supported by grants from the Knut and Alice Wallenberg Foundation (KAW) and the Swedish Research Council (VR).

## Author contributions
P.B. conceived and supervised the research, M.B. prepared samples and performed the experiments, P.B. and M.B. evaluated the data and wrote the manuscript.

## Additional information

**Competing interests:** The authors declare no competing interests.

