## [Peer Review File · Nature Communications]

Reviewers' comments:

Reviewer #1 (Remarks to the Author):

The manuscript by Bjork and Brzezinski reports data on spectral changes in oxidative part of the catalytic cycle of cytochrome c oxidase in mitochondrial membranes dependently on the electric potential generated by ATPase. From two electrogenic state transitions, P to F and F to O, only the later was found to be influenced by the electric potential. Authors made a conclusion that the generation of the electric potential by ATPase across the membrane of submitochondrial vesicles results in retaining of one electron in the binuclear site in the intermediate state FR (heme a oxoferryl and reduced CuB) and, thus, incomplete oxidation of the enzyme. This is of high interest and could help for the further revealing of the molecular mechanism of proton pumping oxidase. However, the statement is very strong; to prove it some control experiments are required. In addition, some clarifications are needed.

The major comments:

1. The aim of this study is to find out the effect of the electric potential on turnover of mitochondrial cytochrome c oxidase. It was reported many times that the electrochemical proton gradient slowed it by a factor (RC) of 5-20. Therefore, it is widely accepted that the value of RC can be considered as a marker of the electric potential across the membrane of SMP. However, in this study determined RC is very low, 1.3, what indicates the negligible amplitude of $\Delta\mu\text{H}^+$.
2. The amplitude of the electric potential measured by Oxonol VI, Fig.2 is extremely low (e.g., an order of magnitude lower than that in observed by Masuya et al, *Biosci Biotechnol Biochem.* 2016 80(8):1464-9 and Kiehl and Hanstein 1984 *BBA* 766:375-383). The transient strong initial increase was not observed previously and not explained. Moreover, it is not eliminated by valinomycin, which should completely dissipate potential in SMP at used KCl concentration. The steady-state potential level is constant for 700 s what contradicts data on Fig.3C: the ratio ATP/SMP in experiments in Fig. 2 (ATP 0.16 mM, SMP 0.1 mg/ml) and 3C (ATP 25 mM, SMP 1.6 mg/ml, both after mixing) is similar, nevertheless "at 600 s ... ATP was fully exhausted" in 3C (page 6 first paragraph) but no potential decay is observed in Fig. 2. The same applies to ΔpH measurements by ACMA: no signal relaxation in 700 s. The control additions of CCCP or gramicidin to dissipate electric potential and nigericin, or gramicidin, or CCCP to dissipate ΔpH are necessary to exclude artificial absorbance changes of the probes.
3. According to authors' conclusion generated by ATPase potential leads to incomplete oxidative half of turnover and, consequently, impaired proton uptake to form H_2O . Without $\Delta\psi$, the time constant of all transitions from PR to O is approx. 5 ms. In the presence of $\Delta\psi$, no kinetics of the PR to O transition was observed up to 60 ms, but it was only suggested that the time constant should be >10 ms what is not informative. Authors claim that a steady state is established after oxidation of Cyt_cO. Is it due to re-reduction with ascorbate? Is it established already at 10 ms? It would be very important to find out when the oxidation of the enzyme in the presence of $\Delta\psi$ is complete and whether the deceleration of the full turnover by $\Delta\mu\text{H}^+$ is caused by the suggested retention of the electron in FR state.
4. The grey line in Fig. 3A (445 nm) is suggested to reflect absorbance changes at the higher electric potential than the black line. The effect is significant, 50%. The same should be indicated in Fig.3B (605 nm).

Minor comments:

1. Page 5, third paragraph: "The time constant of the reaction is typical ~ 1 ms in detergent solution 29, but in the SMPs it was slowed to ~ 5 ms (Figure 3AB), as observed previously for membrane-bound Cyt_cO 30,31" In the Reference 30 this constant was 0.5 ms.
2. Page 6, second paragraph: "The absorbance changes associated with the PR to F reaction are small 7 and could not be resolved here with the scattering SMPs". It is not clear why these changes are small because in the scheme, Fig. 4, upon the PR to F transition the electron is transferred from CuA to heme a, therefore the reduction of heme a should be observed. Also,

there is some contradiction: if the absorbance changes associated with PR to F transition could not be observed how it was concluded that this reaction is not affected by the membrane potential? It should be clarified.

3. Abstract: "The data indicate that proton uptake to the catalytic site and presumably proton translocation were impaired by the potential, but electron transfer was not affected." This phrase is not formulated well since only the electron transfer $\pm \Delta\psi$ was followed in the paper.

Reviewer #2 (Remarks to the Author):

This paper approaches an important question, which is how the membrane potential influences the rates of the individual electron and proton transfer reactions in Cytochrome c Oxidase (CcO). This is an interesting question on many levels. CcO is the terminal electron acceptor, moves 2 charges across the membrane per reduction event building the potential. When the mitochondria has reached a high membrane potential the difficulty for CcO continue to push charges against the gradient can lead to the overall multi-protein electron transfer chain backing up leading to production of reactive oxygen species. Thus, this work has the potential to see the limits of charging the mitochondria.

The question addressed here also is important for our understanding of CcO basic function. The way in which each individual step in the reaction cycle responds to the back pressure of a large gradient can provide information about the thermodynamics of each step. One important outcome is the novel separation of two stages of the F to O transition. Brezinski contributed to methods that helped dissect the P to F reactions and the separating of electron and proton transfers (which happen simultaneously without the membrane potential) in the new work study should prove to be as useful in the analysis of the reaction.

However, while the paper addresses an important problem which is quite difficult to access experimentally. this paper lacks key information, in particular there is no estimate what is the membrane potential that is generated by the ATPase. Without some estimate of this the kinetic changes lack context.

Page 3 (top paragraph): The description of earlier results is more confusing than helpful. Last paragraph before results, it might be useful to note at the outset that heme a to heme a₃ is not a transmembrane electron transfer so should not be affected by the membrane potential.

Page 4. The membrane potential is the main 'tool' used in these experiments. There needs to be a bit more information given. First the estimated potential and ΔpH should be provided. Also, some brief explanation for why the membrane potential decays after a maximum is achieved and why is the building up of membrane potential much faster than that of the ΔpH should be provided.

Page 5. What does it mean that proton uptake to the catalytic site is 'linked' to the slower electron transfer from CuA to heme a. In the fully reduced protein the electron transfer in the Pr>F transition (Heme a to the catalytic site) is not electrogenic while the electron transfer that rereduces heme a by electron transfer from CuA will be electrogenic.

The importance of the electron transfers only moving partially across the membrane while proton pumping transfers across the total membrane (and is thus should feel the full impact of the potential) is not explicitly considered.

Page 6. The ΔG for electron transfer from CuA to Heme a is generally considered to be small. This makes it more surprising that this step is not affected by the transmembrane potential.

Fig 3c. It seems as if the a line should not be interpolated from the first point to the second in a manner that suggests the rate of this process. There should be error bars on the points.

Fig 4 should indicate which steps are changed with the potential.

Reviewer #3 (Remarks to the Author):

This manuscript reports on a very interesting and important observation related to the mechanistic principles of operation of one of the crucial enzyme of mitochondrial electron transport chain. As such, in my view it has potential for publication in Nature Communications. However, my major reservation is that in its present form it lacks broader perspective, which is supposedly required for publication in those types of journals. The discussion sticks just to the results without attempts to go beyond and consider the importance/significance of this result from physiological and cellular perspective. How this might relate to the mechanism of regulation of entire electron transport chain (mitochondrial respiration) etc. How general this principle might be, for example in relation to other proton pumping complexes, bacterial vs mitochondrial? These are just examples of general issues which, if addressed, would make the paper more appealing to a general audience.

On the other hand, the title seems too general; it would be more appropriate to specify which complex of electron transport chain is targeted in this paper, for example by „Control of transmembrane charge transfer by at the level of cytochrome c oxidase by the membrane potential“.

Overall, the paper is very concisely written, maybe even too concisely, which makes understanding of some parts difficult (especially for the nonspecialist reader).

In Introduction the reaction sequence is difficult to follow without any scheme – but the scheme to which the sequence refers to is in Figure 4 (end of the ms). This is why Figure 4 should appear as number 1 or 2 just after Figure The experimental system. Also, in this figure it would be beneficial to put the time constants of the transitions from state R to A and Pr to F etc.

For non specialists it is not clear what happens after the state O when one H₂O is ready to be released but the second water is deprotonated in the form of hydroxyl anion. Is there another step of proton uptake to protonate OH⁻ ion to water and is this additional step also affected by the membrane potential?

Is there a change in the rate of absorbance change in the presence or absence of membrane potential, since in the figure 3A it looks like that the final level of heme a+a₃ is changed but not necessary the rates. If the rates are not affected what would be a possible explanation?

Other points:

1. The last sentence in Introduction is not clear and should be made more comprehensive as the reviewer does not understand “suggestion of a mechanistic principle”.
2. inner mitochondria membrane or mitochondrial membrane?
3. Does transmembrane potential refers to electric potential $\Delta\psi$ and proton concentration to ΔpH ? This should be clarified.
4. Does the dye oxonol VI incorporates into the membrane or is water soluble? It would be informative to explain a bit about the fluorescent dyes: oxonol VI i ACMA, how they work etc?
5. It will be informative to mention that valinomycin abolishes the electric membrane potential but not ΔpH .
6. Page 8 “injection” of electron - injection should be replaced with other sentence.
7. Sentence „Part of the free energy released by this electron transfer” should rather state „Part of the free energy released during this electron transfer process”
8. It would be informative to show (and better explain) the data used to determine and report on

the orientation of the SMP used in this study.

Reviewers' comments in blue, our answers in black.

Reviewer #1

The major comments:

1. The aim of this study is to find out the effect of the electric potential on turnover of mitochondrial cytochrome c oxidase. It was reported many times that the electrochemical proton gradient slowed it by a factor (RC) of 5-20. Therefore, it is widely accepted that the value of RC can be considered as a marker of the electric potential across the membrane of SMP. However, in this study determined RC is very low, 1.3, what indicates the negligible amplitude of $\Delta\mu\text{H}^+$.

The relatively high RC values of 5-20 are typically obtained with cytochrome c oxidase reconstituted in liposomes, i.e. the main constituent of the membrane is lipids (with a low density of proteins). We did quote these data in the introduction section without pointing out that the conditions were not the same as those in the current study. The text has been modified.

Earlier studies with sub-mitochondrial particles, which contain a large fraction of membrane proteins, yield lower RC values. For example, in the study of Grivennikova *et al.* (*FEBS Lett.* 347 (1994) 243) a value of 3.5 was obtained when using NADH as a substrate. In a study by Lee, Ernster and Chance (*Eur. J. Biochem.* 8 (1969) 153) values in the range 2.3-3.3 or 1.4-2.5 were obtained, when using substrate NADH or succinate, respectively. We used the latter substrate in our experiments. In addition, the authors of both cited studies noted that the use of oligomycin was necessary to obtain these RC values. In our study we used ATP-synthase to establish and maintain the membrane potential, which is the reason oligomycin was excluded. On the other hand, because in our study the ATP synthase pumped protons during the measurement, proton leaks via the pumping pathway are not relevant. The authors of the latter study noted that "the particles possessed a certain extent of respiratory control even in the absence of oligomycin" and these are the values that we have measured.

We repeated the RC measurement 10 times and obtained an average of 1.30 with a SD of 0.04, i.e. the value is significantly larger than unity (but we agree that it is small).

Comments on this issue have been added to the revised manuscript.

2. The amplitude of the electric potential measured by Oxonol VI, Fig.2 is extremely low (e.g., an order of magnitude lower than that in observed by Masuya *et al.*, *Biosci Biotechnol Biochem.* 2016 80(8):1464-9 and Kiehl and Hanstein 1984 *BBA* 766:375-383).

The earlier measurements were performed using plastic cuvettes. It turns out that the plastic material appears to bind some of the dye, which explains the lower amplitude in the measurements. We have now repeated all measurements with quartz cuvettes and obtained about the same values as those in the cited papers. The figure has been replaced with the new data.

3. The transient strong initial increase was not observed previously and not explained. Moreover, it is not eliminated by valinomycin, which should completely dissipate potential in SMP at used KCl concentration.

The initial increase originates from mixing. Most likely, it was also linked to the binding of the dye to the cuvette (see point 2). The new data show essentially no initial transient increase.

Addition of valinomycin was done at 80 s, i.e. it was not present from start. The time points for the additions are indicated in the revised version of the manuscript.

4. The steady-state potential level is constant for 700 s what contradicts data on Fig.3C: the ratio ATP/SMP in experiments in Fig. 2 (ATP 0.16 mM, SMP 0.1 mg/ml) and 3C (ATP 25 mM, SMP 1.6 mg/ml, both after mixing) is similar, nevertheless "at 600 s ... ATP was fully exhausted" in 3C (page 6 first paragraph) but no potential decay is observed in Fig. 2. The same applies to ΔpH measurements by ACMA: no signal relaxation in 700 s.

This is a very good point. However, the reviewer cites the ATP and SMP concentrations used in the experiments in panels A and B of Fig. 3. The experiment in panel 3C was performed with different concentrations because in this experiment the ATP and SMP solutions were incubated before mixing. These concentrations are 1 mM ATP and 5.5 mM SMP (the SMP concentration given in the first version of the manuscript was incorrect).

5. The control additions of CCCP or gramicidin to dissipate electric potential and nigericin, or gramicidin, or CCCP to dissipate ΔpH are necessary to exclude artificial absorbance changes of the probes.

These controls have been added to Fig. 2.

6. According to authors' conclusion generated by ATPase potential leads to incomplete oxidative half of turnover and, consequently, impaired proton uptake to form H_2O . Without $\Delta\psi$, the time constant of all transitions from PR to O is approx. 5 ms. In the presence of $\Delta\psi$, no kinetics of the PR to O transition was observed up to 60 ms, but it was only suggested that the time constant should be >10 ms what is not informative. Authors claim that a steady state is established after oxidation of Cyt c O. Is it due to re-reduction with ascorbate? Is it established already at 10 ms? It would be very important to find out when the oxidation of the enzyme in the presence of $\Delta\psi$ is complete and whether the deceleration of the full turnover by $\Delta\mu\text{H}^+$ is caused by the suggested retention of the electron in FR state.

We agree that this is an important point and we tried to address it. The main problem is that the conditions required to carry out the experiment (i.e. the presence of ascorbate that is used to prepare the reduced Cyt c O), also lead to re-reduction of the enzyme upon oxidation. The turnover is not initiated directly at 10 ms. However, heme α becomes re-reduced, which is seen as an increasing slope in the absorbance changes at 605 nm at times >20 ms. Consequently, when state FR decays to state O, the electron at heme α is presumably transferred to the oxidized catalytic site. We did try to optimize the ascorbate concentration (i.e. to lower the ascorbate concentration), but it turns out that

because of the delay in the decay of state FR, there is sufficient time for re-reduction of heme *a* also at lower ascorbate concentrations.

Theoretically one could perform the experiment using small amount (less than the O₂ concentration after mixing) of dithionite to reduce the Cyt_cO. The excess dithionite would be consumed by oxygen upon mixing and the remaining fraction of O₂ would eventually oxidize the Cyt_cO. Unfortunately, we found that the use of dithionite is incompatible with the use of ATP.

7. The grey line in Fig. 3A (445 nm) is suggested to reflect absorbance changes at the higher electric potential than the black line. The effect is significant, 50%. The same should be indicated in Fig.3B (605 nm).

These data were excluded for technical reasons. As indicated in the legend, the data shown in panels A and B in Fig. 3 are averages of 10-20 traces. The data at the higher electric potential (Fig. 3C) could only be collected as single traces at specific time points. This was possible at 445 nm, but at 605 nm the signals are typically a factor of ~5 smaller (with a similar or slightly larger noise level). Consequently, we did not include a trace in Fig. 3B. However, we repeated the experiments at 605 nm, as suggested by the reviewer, and present the amplitudes of the F → O component, determined from the average absorbance levels before and after the component. These data have been added to panel C in Fig. 3 and contain all the information requested by the reviewer.

Minor comments:

8. Page 5, third paragraph: “The time constant of the reaction is typical ~1 ms in detergent solution 29, but in the SMPs it was slowed to ~5 ms (Figure 3AB), as observed previously for membrane-bound Cyt_cO 30,31” In the Reference 30 this constant was 0.5 ms.

Corrected

9. Page 6, second paragraph: “The absorbance changes associated with the PR to F reaction are small 7 and could not be resolved here with the scattering SMPs”. It is not clear why these changes are small because in the scheme, Fig. 4, upon the PR to F transition the electron is transferred from Cu_A to heme *a*, therefore the reduction of heme *a* should be observed. Also, there is some contradiction: if the absorbance changes associated with PR to F transition could not be observed how it was concluded that this reaction is not affected by the membrane potential? It should be clarified.

This process was not explained well. Absorbance changes associated with the P_R → F reaction are very small (at the maximum wavelength, 580 nm, a factor of ~5 smaller than those associated with heme *a* reduction at 605 nm). The reaction is typically linked in time to a shift in the equilibrium between Cu_A and heme *a*, which yields reduction of the latter. The Cu_A - heme *a* equilibrium constant during the P_R → F reaction varies for different oxidases and with experimental conditions. In the SMPs studied here the fraction reduced heme *a* was smaller than for the bovine Cyt_cO in detergent, i.e. in the

SMPs the main fraction of the electron remained at Cu_A . We have modified the text to avoid this confusion.

We have also modified the conclusion discussing the effect on the $\text{P}_R \rightarrow \text{F}$ reaction.

10. Abstract: "The data indicate that proton uptake to the catalytic site and presumably proton translocation were impaired by the potential, but electron transfer was not affected." This phrase is not formulated well since only the electron transfer $\pm \Delta\psi$ was followed in the paper.

We have modified the sentence: "The data indicate that a reaction step that involves proton uptake to the catalytic site..".

Reviewer #2:

This paper approaches an important question, which is how the membrane potential influences the rates of the individual electron and proton transfer reactions in Cytochrome c Oxidase (CcO). This is an interesting question on many levels. CcO is the terminal electron acceptor, moves 2 charges across the membrane per reduction event building the potential. When the mitochondria has reached a high membrane potential the difficulty for CcO continue to push charges against the gradient can lead to the overall multi-protein electron transfer chain backing up leading to production of reactive oxygen species. Thus, this work has the potential to see the limits of charging the mitochondria.

The question addressed here also is important for our understanding of CcO basic function. The way in which each individual step in the reaction cycle responds to the back pressure of a large gradient can provide information about the thermodynamics of each step. One important outcome is the novel separation of two stages of the F to O transition. Brezinski contributed to methods that helped dissect the P to F reactions and the separating of electron and proton transfers (which happen simultaneously without the membrane potential) in the new work study should prove to be as useful in the analysis of the reaction.

1. However, while the paper addresses an important problem which is quite difficult to access experimentally. this paper lacks key information, in particular there is no estimate what is the membrane potential that is generated by the ATPase. Without some estimate of this the kinetic changes lack context.

We attempted to calibrate the potential by using a range of different K^+ concentrations on the inside and outside of the SMPs in combination with valinomycin, but, unfortunately, the results were not meaningful. Instead, we repeated the potential measurements with the SMPs and then used empty lipid vesicles with the same size as that of the SMPs, and using identical conditions. Using these data the membrane

potential in the SMPs was estimated to be ~ 100 mV. The calibration curve is presented in a "supplementary information" file.

2. Page 3 (top paragraph): The description of earlier results is more confusing than helpful.

Last paragraph before results, it might be useful to note at the outset that heme a to heme a₃ is not a transmembrane electron transfer so should not be affected by the membrane potential.

Both sections have been rewritten. We now also present a reaction scheme in the introduction section (new Fig. 1B).

3. Page 4. The membrane potential is the main 'tool' used in these experiments. There needs to be a bit more information given. First the estimated potential and ΔpH should be provided. Also, some brief explanation for why the membrane potential decays after a maximum is achieved and why the building up of membrane potential is much faster than that of the ΔpH should be provided.

We have done a calibration of the membrane potential (see point 1 above). Upon addition of nigericin (transforms ΔpH to $\Delta\Psi$ under the conditions of our experiment) the electrical potential increased as shown in the new version of Fig. 2A. We estimated the fraction ΔpH after addition of ATP and found it to be ~ 0.4 pH units. We have added this information to the text.

Establishing the electrical potential is much faster than formation of a proton concentration gradient because the former requires transfer of only a few charges across the membrane (charging of a spherical lipid-membrane capacitor), while formation of a concentration (pH) gradient requires a large number of turnovers by the ATP synthase. The decay in the membrane potential is presumably caused by equilibrium of proton pumping and all possible ion leaks being established across the membrane. Both these points are addressed in the revised version of the manuscript.

Page 5. What does it mean that proton uptake to the catalytic site is 'linked' to the slower electron transfer from Cu_A to heme a. In the fully reduced protein the electron transfer in the Pr⁻→F transition (Heme a to the catalytic site) is not electrogenic while the electron transfer that rereduces heme a by electron transfer from Cu_A will be electrogenic.

The importance of the electron transfers only moving partially across the membrane while proton pumping transfers across the total membrane (and is thus should feel the full impact of the potential) is not explicitly considered.

In the revised version we explain the order of electron- and proton-transfer reactions in the Pr⁻→F reaction. The term "linked" has been changed to "occurs over the same time". The effect of membrane potential on the different steps has also been explained in more detail. We also mention discuss the electron transfer from Cu_A to heme a.

Page 6. The ΔG for electron transfer from Cu_A to Heme a is generally considered to be small. This makes it more surprising that this step is not affected by the transmembrane potential.

The explanation is as follows. At the time when P_R is formed, Cu_A is reduced and heme a is oxidized. During the $Pr\text{--}F$ reaction the Cu_A - heme a equilibrium is shifted towards heme a , which is observed as a fractional reduction of heme a . The extent of heme a reduction varies between oxidases from different species and experimental conditions. For example, with the detergent-solubilized bovine heart Cyt c O it is shifted more towards heme a than for the *R. sphaeroides aa₃* Cyt c O. In the sub-mitochondrial particles studied in the current work, the Cu_A - heme a electron equilibrium in the absence of a membrane potential was shifted toward Cu_A , i.e. a very small fraction heme a was reduced. The effect of the applied membrane potential (positive on the Cu_A side of the Cyt c O) is to shift the electron equilibrium even more toward Cu_A , which explains the absence of a potential effect on this electron transfer.

Fig 3c. It seems as if the a line should not be interpolated from the first point to the second in a manner that suggests the rate of this process. There should be error bars on the points.

The line has been removed. We mention in the figure legend that the error bars are of approximately the same size as the symbols.

Fig 4 should indicate which steps are changed with the potential.

Fig.4 (new Fig. 1B) has been modified to include this information

Reviewer #3

This manuscript reports on a very interesting and important observation related to the mechanistic principles of operation of one of the crucial enzyme of mitochondrial electron transport chain. As such, in my view it has potential for publication in Nature Communications. However, my major reservation is that in its present form it lacks broader perspective, which is supposedly required for publication in those types of journals. The discussion sticks just to the results without attempts to go beyond and consider the importance/significance of this result from physiological and cellular perspective. How this might relate to the mechanism of regulation of entire electron transport chain (mitochondrial respiration) etc. How general this principle might be, for example in relation to other proton pumping complexes, bacterial vs mitochondrial? These are just examples of general issues which, if addressed, would make the paper more appealing to a general audience.

These issues are mentioned and discussed in the revised version of the manuscript.

On the other hand, the title seems too general; it would be more appropriate to specify which complex of electron transport chain is targeted in this paper, for example by „Control of transmembrane charge transfer by at the level of cytochrome c oxidase by the membrane potential”.

The title has been modified.

Overall, the paper is very concisely written, maybe even too concisely, which makes understanding of some parts difficult (especially for the nonspecialist reader).

In Introduction the reaction sequence is difficult to follow without any scheme – but the scheme to which the sequence refers to is in Figure 4 (end of the ms). This is why Figure 4 should appear as number 1 or 2 just after Figure The experimental system. Also, in this figure it would be beneficial to put the time constants of the transitions from state R to A and Pr to F etc.

We removed Fig.4 and added the scheme as panel B in Fig.1. As the reviewer points out this makes it easier to follow the introduction and discussion.

For non specialists it is not clear what happens after the state O when one H₂O is ready to be released but the second water is deprotonated in the form of hydroxyl anion. Is there another step of proton uptake to protonate OH⁻ ion to water and is this additional step also affected by the membrane potential?

The information has been added to the figure.

Is there a change in the rate of absorbance change in the presence or absence of membrane potential, since in the figure 3A it looks like that the final level of heme a+a₃ is changed but not necessary the rates. If the rates are not affected what would be a possible explanation?

This observation is explained in detail in the revised version

Other points:

1. The last sentence in Introduction is not clear and should be made more comprehensive as the reviewer does not understand “suggestion of a mechanistic principle”.

The sentence has been rewritten.

2. inner mitochondria membrane or mitochondrial membrane?

Fixed.

3. Does transmembrane potential refers to electric potential $\Delta\psi$ and proton concentration to ΔpH ? This should be clarified.

We have modified the text.

4. Does the dye oxonol VI incorporates into the membrane or is water soluble? It would be informative to explain a bit about the fluorescent dyes: oxonol VI i ACMA, how they work etc?

We now mention references to a detailed description of these dyes.

5. It will be informative to mention that valinomycin abolishes the electric membrane potential but not ΔpH .

This information has been added to the revised version of the manuscript.

6. Page 8 “injection” of electron - injection should be replaced with other sentence.

Done

7. Sentence „Part of the free energy released by this electron transfer” should rather state „Part of the free energy released during this electron transfer process”

Done

8. It would be informative to show (and better explain) the data used to determine and report on the orientation of the SMP used in this study.

The data have been added to "supplementary information" (new file).

REVIEWERS' COMMENTS:

Reviewer #1 (Remarks to the Author):

After the revision, the paper was significantly improved. The potential measurements no longer cause doubt (although it is not clear why valinomycin was added with FCCP to abolish $\Delta\mu\text{H}^+$; since ATPase is the proton pump FCCP alone completely dissipates created $\Delta\mu\text{H}^+$).

However, the article would benefit from some minor improvements and clarification.

1. Since the PR \rightarrow F transition was not found to be $\Delta\psi$ dependent, the authors came to the very important conclusion that there is no transmembrane H^+ transfer upon this transition. Does it mean that the stoichiometry $\text{H}^+:\text{e}^-$ changes in dependence on the electric potential? The loss of proton translocation upon the fast (approx. 200 μs) phase of the reaction contradicts direct electrometry measurements proving that the PR \rightarrow F transition in the aa3-type oxidases is coupled to transfer of about 1.3–1.6 charges across the membrane (Belevich et al. PNAS 2010. 107 (43) 18469., Belevich et al. Nature 2006, 440 829). That should be discussed.

2. Fig. 2 inset A: it would be better to limit the scale to approx. 10 s and clearly mark the point 0.8 s, when the reaction was initiated by the laser flash.

3. The panels A and B in Figure 2 do not indicate the significant decrease of proton electrochemical gradient up to at least 500 s, because the drop in the electric potential corresponds to ΔpH generation thus, $\Delta\mu\text{H}^+$ hardly changes. Therefore, the conclusion in line 220 "At times >280 s the amplitude increased again to reach $\sim 100\%$ at ~ 600 s, presumably when the electrochemical potential decreased" cannot be solid.

4. There is a contradiction in lines: 60 "Each electron transfer to the catalytic site is linked to proton pumping", 183: "The PR \rightarrow F reaction is associated with pumping of one proton across the membrane" and line 100 "state FR was formed before proton uptake and pumping".

5. Line 333. "Consequently, it would be difficult to regulate Cyt_cO activity by altering the driving force for electron transfer by the external electric potential. " However, it is well shown than Cyt_cO activity is strongly depressed by electric potential created by itself: in proteoliposomes, RC could be over 15.

.

6. The phrase in line 289 is not clear "This shift yields absorbance changes, for example, at 445 nm and at 605 nm, which are larger than that associated with the PR \rightarrow F reaction itself 20. "

Reviewer #2 (Remarks to the Author):

The authors have generally addressed my questions and comments. My general view is that this is difficult to obtain but important information and is appropriate for Nature as the results have impacts on the combined electron transfer chain. My one remaining question is on their estimate of the transmembrane potential is 100 mV. The $\Delta\mu\text{H}$ is generally quoted as being closer to 180-200 mV. The conclusions thus seem to be there are changes in reaction rates occurring at potentials significantly lower than the running potential for the cell. This should be addressed and the estimates of their experimental potential and found changes in rate vs what would be expected in a mitochondria at

running at full capacity should be included in the main text.

Reviewer #3 (Remarks to the Author):

In general the authors amended the manuscript in several parts according to the reviewer's comment which made the manuscript easier to follow for non-specialist. The improved parts encompass the scheme in fig. 1 which now clearly describes the steps of the reactions that are monitored in the experiments presented in the manuscript and other figures. However a discussion on a possible physiological significance and effect of the decrease in pumping of protons in the presence of the membrane potential on the whole respiratory chain is only mentioned and no „speculation“ on the role of this effect was provided by authors. Simply one sentence was added in lines 317-328. This is not a plea but in reviewers's opinion such discussion would be beneficial for setting a broader perspective of the findings presented in the text.

Also I did not find the discussion on the possible change in the rate of 5-ms component (grey line in fig 3A) in the presence of the membrane potential i.e. comparison to red and black traces with grey . Authors still discuss only the yield and not the rate. So the question is - does the membrane potential affects the yield of electron of the rate of electron transfer, or both? This should be clearly stated before publication.

I would also suggest to clearly state that 5-ms component is the F to O transition and it should be mentioned before introducing this term (somewhere at line 186).

REVIEWERS' COMMENTS:

Reviewers' comments in blue, our answer in black

Reviewer #1:

After the revision, the paper was significantly improved. The potential measurements no longer cause doubt (although it is not clear why valinomycin was added with FCCP to abolish $\Delta\mu_{H^+}$; since ATPase is the proton pump FCCP alone completely dissipates created $\Delta\mu_{H^+}$). However, the article would benefit from some minor improvements and clarification.

1. Since the PR \rightarrow F transition was not found to be $\Delta\psi$ dependent, the authors came to the very important conclusion that there is no transmembrane H^+ transfer upon this transition. Does it mean that the stoichiometry $H^+:e^-$ changes in dependence on the electric potential? The loss of proton translocation upon the fast (approx. 200 μ s) phase of the reaction contradicts direct electrometry measurements proving that the PR \rightarrow F transition in the aa3-type oxidases is coupled to transfer of about 1.3–1.6 charges across the membrane (Belevich et al. PNAS 2010. 107 (43) 18469., Belevich et al. Nature 2006, 440 829). That should be discussed.

We fully agree with the reviewer on this point. Our description of the data found in the literature was unclear. We have added two sentences and added the two references mentioned above.

2. Fig. 2 inset A: it would be better to limit the scale to approx. 10 s and clearly mark the point 0.8 s, when the reaction was initiated by the laser flash.

We have modified the time scale and added an arrow indicating the 0.8 s time point.

3. The panels A and B in Figure 2 do not indicate the significant decrease of proton electrochemical gradient up to at least 500 s, because the drop in the electric potential corresponds to Δ pH generation thus, $\Delta\mu_{H^+}$ hardly changes. Therefore, the conclusion in line 220 "At times >280 s the amplitude increased again to reach ~100 % at ~600 s, presumably when the electrochemical potential decreased" cannot be solid.

We have clarified the conclusion. It is true that the drop in the electrical potential is linked in time to generation of Δ pH, but we could not quantify the latter. Furthermore, there is a slow decrease in $\Delta\Psi$ at a time when Δ pH has reached a constant value at ~300 s.

4. There is a contradiction in lines: 60 "Each electron transfer to the catalytic site is linked to proton pumping", 183: "The PR \rightarrow F reaction is associated with pumping of one proton across the membrane" and line 100 "state FR was formed before proton uptake and pumping".

We meant to say that on average one proton is pumped per electron transferred to O₂. This point has been clarified in the introduction.

5. Line 333. "Consequently, it would be difficult to regulate Cyt cO activity by altering the driving force for electron transfer by the external electric potential." However, it is well shown that Cyt cO activity is strongly depressed by electric potential created by itself: in proteoliposomes, RC could be over 15.

The sentence has been modified.

6. The phrase in line 289 is not clear "This shift yields absorbance changes, for example, at 445 nm and at 605 nm, which are larger than that associated with the PR → F reaction itself 20."

The sentence has been modified.

Reviewer #2:

The authors have generally addressed my questions and comments. My general view is that this is difficult to obtain but important information and is appropriate for Nature as the results have impacts on the combined electron transfer chain. My one remaining question is on their estimate of the transmembrane potential is 100 mV. The $\Delta\mu_H$ is generally quoted as being closer to 180-200 mV. The conclusions thus seem to be there are changes in reaction rates occurring at potentials significantly lower than the running potential for the cell. This should be addressed and the estimates of their experimental potential and found changes in rate vs what would be expected in a mitochondria at running at full capacity should be included in the main text.

We have added a comment on the differences in transmembrane potentials and that the observed effects occur already at lower potentials than that found *in vivo*.

Reviewer #3:

In general the authors amended the manuscript in several parts according to the reviewer's comment which made the manuscript easier to follow for non-specialist. The improved parts encompass the scheme in fig. 1 which now clearly describes the steps of the reactions that are monitored in the experiments presented in the manuscript and other figures. However a discussion on a possible physiological significance and effect of the decrease in pumping of protons in the presence of the membrane potential on the whole respiratory chain is only mentioned and no „speculation" on the role of this effect was provided by authors. Simply one sentence was added in lines 317-328. This is not a plea but in reviewers's opinion

such discussion would be beneficial for setting a broader perspective of the findings presented in the text.

We have rearranged three paragraphs at the end of the Discussion and extended the last one with some speculation.

Also I did not find the discussion on the possible change in the rate of 5-ms component (grey line in fig 3A) in the presence of the membrane potential i.e. comparison to red and black traces with grey . Authors still discuss only the yield and not the rate. So the question is - does the membrane potential affects the yield of electron of the rate of electron transfer, or both? This should be clearly stated before publication.

The membrane potential affects both the amplitude and the rate; the amplitude of the 5-ms component ($F \rightarrow F_R$) because is smaller because the rate of the $F_R \rightarrow O$ reaction is slowed. We have added an explanation.

I would also suggest to clearly state that 5-ms component is the F to O transition and it should be mentioned before introducing this term (somewhere at line 186).

Done.